# Detection of Direct Sun Glare on Drivers from Point Clouds

Silvia María González-Collazo *, Pablo del Río-Barral ⓘ, Jesús Balado ⓘ and Elena González ⓘ

GeoTECH Group, CINTECX, Universidade de Vigo, 36310 Vigo, Spain; rio.barral.pablo@uvigo.es (P.d.R.-B.);
jbalado@uvigo.es (J.B.); elena@uvigo.es (E.G.)
* Correspondence: silvgonzalez@uvigo.es

**Abstract:** Sunlight conditions can reduce drivers' visibility, which is a safety concern on road networks. This research introduces a method to study sun glare incidence in road environments. Sun glare areas during daylight hours are automatically detected from mobile laser scanning (MLS) and aerial laser scanning (ALS) point clouds. The method comprises the following steps. First, the Sun's position (solar altitude and azimuth) referring to a location is calculated. Second, the incidence of sun glare with the user's angle of vision is analyzed based on human vision. Third, sun ray intersections with near obstacles (vegetation, building, etc.) are calculated utilizing MLS point clouds. Finally, intersections with distant obstacles (mountains) are calculated utilizing ALS point clouds. MLS and ALS data are processed in order to combine both data types, remove outliers, and optimize computational time for intersection searches (point density reduction and Delaunay triangulation). The method was tested on two real case studies, covering roads with different bearings, slopes, and surroundings. The combination of MLS and ALS data, together with the solar geometry, identify areas of risk for the visibility of drivers. Consequently, the proposed method can be utilized to reduce sun glare, implementing warnings in navigation systems.

**Keywords:** visibility; solar incidence; road geometry; safety; shadow; road environment





## 1. Introduction

In modern societies and developed countries, urban and interurban road networks constitute a fundamental infrastructure for the movement of people and goods. A large proportion of citizens employ the road network on a daily basis to communicate with other places such as factories, hospitals, schools, and stores. Based on data from the Spanish Ministry of Transport, Mobility, and Urban Agenda [1], approximately 14,000 vehicles per day travel on the national road network. Furthermore, this high occupancy has led to an increase in road kilometers over the last 50 years. European statistics show that the number of vehicles has risen in most European countries since 2010 [2]. In 2019, there were more than 103,000,000 passenger cars and more than 13,000,000 goods vehicles considering Spain, Germany, and France. Dealing with the growing creation and use of roads, searching for safer roads and developed infrastructures becomes a crucial research line. Consequently, driving improvement is sought in redesigning roads and analyzing possible issues which can affect road safety, such as visibility, weather conditions, and road geometry.

Concerning road safety, visibility has been analyzed by many researchers considering weather conditions (rain, snow, and fog), sunlight, sun glare, road geometry, and road obstacles. However, most of these factors have been studied separately and with a statistical approach. Accidents likelihood depending on weather conditions was analyzed by [3–5], concluding adverse weather conditions increase the accident likelihood, as can be expected. The authors of [6] focused their analysis on the degradation of visibility produced by fog. They presented a neural network approach to estimate the visibility range under these conditions using a camera. The authors of [7] examined the impact of reduced visibility under hazy weather conditions on collision risk. They observed overall collision risk and speed variation are higher under hazy weather conditions. In addition, road geometry and

road obstacles influence the visibility of pedestrians and vulnerable road users. The authors of [8] concluded visibility is a relevant factor regarding safety, particularly in intersections which are complex areas due to their visual obstructions.

Sunlight and its incidence on the road are significant features affecting visibility. Sun glare leads to distortion of the road conditions, and in some cases, the driver is blinded by the sunlight. As a result, the driver lacks time to adequately manage all the factors on the road, which leads to road traffic accidents [9]. Sun glare has been an issue for manual driving, however, it is becoming a real problem for autonomous driving as well. Information to be collected by the camera from the area where sun is shining may be lost [10]. The majority of the work studying solar incidence and the Sun's position focuses on energetic efficiency analysis [11–13]; however, the impact of sun glare on road visibility has been studied less extensively. A review of the effect of traffic and different weather conditions on accidents is shown in [5]. According to [14,15], sun glare significantly contributes to collision occurrence, especially at road intersections. Moreover, they observed roads are more affected by sun glare during the main winter months, due to the lower position of the Sun.

Despite there being many articles related to road safety, few authors have studied in depth how sunlight influences road users' visibility, also analyzing road geometry and the obstacles within it. In this article, a method is presented to automatically identify those areas where sun glare occurs from the view of the driver. The method takes advantage of point clouds as geo-referenced 3D information and solar geometry to perform a detailed analysis of the road. The proposed method consists of the following steps. Given the location of the driver on the road, the altitude and azimuth of the Sun are calculated for the desired date and time. Then, the driver's bearing taken from the mobile laser scanning (MLS) acquisition trajectory is contrasted with the orientation of the sun rays. Finally, intersections with near and distant obstacles are searched in MLS and aerial laser scanning (ALS) data, for which it is necessary to process the point clouds to remove outliers, reduce density, and triangulate them. This process can be done spatially along the road as well as temporally for a location throughout the day.

## 2. Related Work
### 2.1. Visibility Analysis

Visibility is a major feature in road safety [16]. Visibility is influenced mainly by road geometry, road obstacles, weather conditions, and sunlight. Several techniques have been proposed to obtain visibility on roads with respect to a given point regarding road geometry and obstacles (buildings, vegetation, and street furniture). The ray-tracing algorithm can detect if a ray from the observer point intersects with some object in the scenario. The authors of [17] defined the line of sight, which starts at the driver's eyes and follows the direction of the trajectory to a target point, checking whether or not there are obstacles between the beginning and the end of the line of sight. They obtained a 2D visibility area from mobile laser scanning (MLS) data and from aerial laser scanning (ALS) data. These areas were compared, showing MLS data was more accurate to close obstacles and ALS data to the farthest ones. The authors of [18] proposed a method to obtain the 3D visibility area in an urban street environment based on LiDAR MLS data. They combined some techniques such as voxelization or the ray-triangle intersection method introduced by [19]. The concept of volume index was presented to compare the visibility of different urban built environments. Their approach was able to measure the volume of the visible space at any place on the street, although it was only applied to street objects. Despite the fact that these analyses show methods to obtain a 2D or 3D visibility area in urban and interurban scenarios, none of them performed research on the effect of sun glare.

Most of the existing literature referring to visibility analyses studied how road geometry and road obstacles obstruct the visibility of the road user, especially the close surroundings of the driver. With regard to previous approaches, in this work, solar incidence is investigated considering road geometry, road obstacles, and vehicle trajectory. An

automatic method for visibility analysis on roads is developed to calculate intersections of sunlight rays with both near obstacles (vegetation, building, road elements, and embankments) and distant obstacles (mountains). The MLS data provides the road geometry and the driver's bearing while ALS data provides the terrain model.

*2.2. Solar Geometry*

The position of the Sun has been extensively studied and it is defined by the equations which will be explained in Section 3. In order to inspect whether sunlight influences the road user or not, it is necessary to know user location, trajectory, and viewing angle. The authors of [20] examined traffic accidents and calculated the Sun's position relative to the first vehicle concerned using the traffic accident database of Chiba Prefecture, Japan. They used the latitude and the longitude of the studied area and the vehicle trajectory to obtain the Sun's position during each accident related to sun glare. They obtained that the number of traffic accidents increases when the Sun is located within 90 degrees of the viewing angle of the first vehicle involved. The same line of research was followed by [21] in the Cape Town road network, in which the incidence of direct sunlight was analyzed. They identified three factors that influence the sun glare effect on roads; geometric design of the road, topography and terrain profile, and the Sun's position. The ArcGIS tool Hillshade was used to model the effect of sun glare exposure. In this way, a terrain analysis was performed, using a digital elevation model (DEM)-based hill-shading to identify roads vulnerable to sun glare conditions. They noticed that the solar illumination in the afternoon periods of the winter and summer solstice showed a greater effect than during the morning period, while the opposite was found in the autumnal and spring equinoxes. A similar procedure was carried out in rural road networks [22]. They proposed a method to analyze the incidence of direct sunlight in the summer season on a specific route, based on a geographic information system (GIS). The route was segmented and afterward, the hill-shade layer was obtained in order to exclude shadowed road segments. They obtained all the road sections that were heavily exposed to sunlight. DEM was used as input data. Consequently, they did not consider road obstacles such as buildings or vegetation, which can also occlude sunlight. GIS-based methods are prevalent to study Sun position. Other authors studied sun glare regions within urban environments based on images, considering the driving direction. The authors of [23] analyzed sun glare regions taking as input data from public images of Google Street View. They examined intersections with buildings, verifying these intersections with images captured in situ at different times. The authors of [24] analyzed sun glare regions using visual explanations for traffic light detection and considering the vehicle orientation. Although the driver's bearing and intersections with obstacles were analyzed, neither of them studied near and distant obstacles in detail that can occlude sunlight.

Georeferenced points clouds are also an information source to locate the Sun's trajectory and calculate intersections between the built environment and sun rays. The authors of [25] and [26] proposed a method to obtain solar radiation and solar envelopes based on information from point cloud data in the built environment. Regarding the previous approaches, in this work, the Sun's position is defined with the solar altitude $\alpha$ and the azimuth angle $\beta$ at a given geographical location, day, and time. Considering together the Sun's position, driver's location, and diver's bearing (road slope and road bearing), the incidence of sunlight within the driver's vision angle is calculated. Moreover, the sunlight trajectory and sun rays relative to a point of study are obtained, therefore, it is possible to analyze whether sun rays intersect with any near (MLS data) or distant obstacle (ALS data) before reaching the point of the road user.

## 3. Method

The method consists of four main processes (Figure 1). First, the Sun's position (solar altitude $\alpha$ and azimuth angle $\beta$) is calculated. Second, the incidence of sunlight is compared with the road user's bearing. Third, intersections of sun rays with near road obstacles from

MLS data are detected. Fourth, intersections of sun rays with distant obstacles from ALS data are detected.

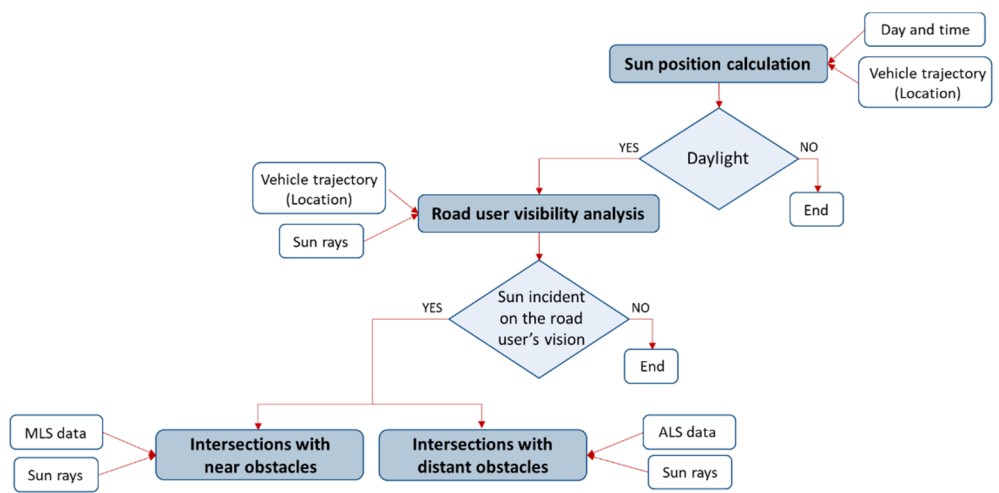

**Figure 1.** Workflow for the proposed method.

### 3.1. Sun Position Calculation

The Sun's position relative to the Earth is mainly determined by the latitude of the geographical area under study and by two angles, the solar azimuth β and the solar altitude α. The Earth rotates around its own axis, known as the polar axis. The equatorial plane divides the Earth into two hemispheres and is perpendicular to the polar axis. Focusing on the translational movement, the ecliptic plane is the plane containing the elliptical orbit of the Earth around the Sun. Due to the polar axis inclination, there is an angle difference between the equatorial plane and the ecliptic plane. This angle is called declination δ and is defined as the angle formed between the ecliptic plane and the equatorial plane. Equation (1) defines the declination for any day of the year, where N denotes the number of elapsed days until the studied day.

$$\delta = 23.45\sin(360 \times (284 + N)/365) \tag{1}$$

Solar altitude α is defined as the elevation of the Sun with respect to the horizon, being the horizontal plane which passes through the observer and is perpendicular to the vertical. Equation (2) defines solar altitude α, where δ is the declination, γ is the latitude of the studied location, and ω the hour angle.

$$\alpha = \mathrm{asin}(\sin(\delta)\sin(\gamma) + \cos(\delta)\cos(\gamma)\cos(\omega)) \tag{2}$$

It should be pointed out that there is a difference between the solar hour and the clock time. The correction of this difference is defined by Equation (3), where $L_{st}$ is the longitude of the country's referenced meridian, C denotes the light saving time correction, ET is a correction factor (Equation (4)) and B (Equation (5)) is related to N (number of elapsed days until the studied day).

$$\mathrm{Solar\_hour} = \mathrm{clock\_time} + ET \pm 4(L_{st} - L_{loc}) + C \tag{3}$$

$$ET = 229.2(0.000075 + 0.001868\cos(B) - 0.032077\sin(B) - 0.014615\cos(2B) - 0.04089\sin(2B)) \tag{4}$$

$$B = (N - 1) \times (360/365) \tag{5}$$

Solar azimuth β is the angle formed by the south direction with the horizontal projection of the straight line connecting the position of the Sun with the observation point. Solar

azimuth β is defined by Equation (6), where δ is the declination angle, ω is the hour angle, and β is the solar altitude.

$$\beta = asin(\cos(\delta) \times \sin(\omega)/\cos(\alpha)) \tag{6}$$

Solar altitude α and azimuth β were analyzed for each season (Figure 2). During the summer solstice, the Sun reaches the highest solar altitude while during the winter solstice it reaches the lowest solar altitude. Regarding each season, the highest solar altitude is reached at 12:00 solar hour.

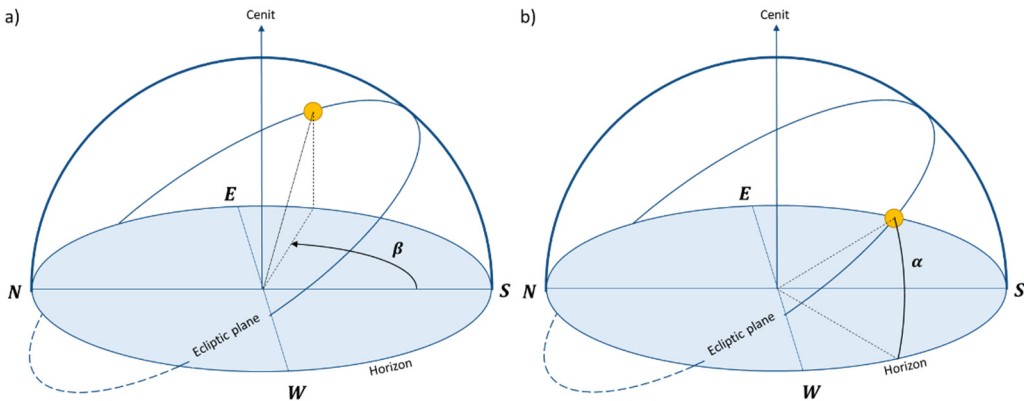

**Figure 2.** The Sun's position during summer solstice: (**a**) azimuth angle β and (**b**) solar altitude α.

Sunrise and sunset hours are calculated for a given location and day, defining the time range in which it is daytime. The solar altitude α and solar azimuth β are then calculated within this time range in order to obtain the Sun's position and calculate the intersections with road obstacles.

### 3.2. Road User's Vision Angle

Direct sun glare is caused by solar incidence in a road user's vision angle. The human vision field is defined by Figure 3. Regarding the horizontal plane, the human vision field range is between 60° and −60°, while on a vertical plane, the human vision field is between 50° and −70°, [27].

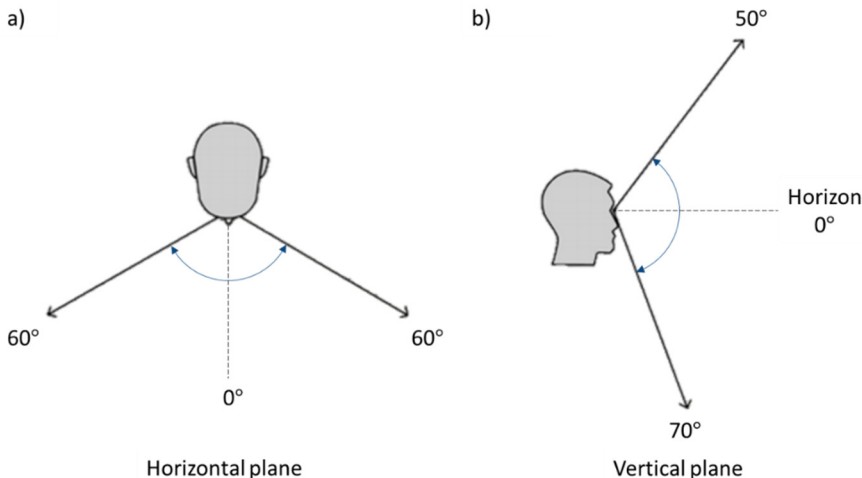

**Figure 3.** Human vision field: (**a**) horizontal plane and (**b**) vertical plane. Figure modified from (Tara et al., 2020).

The azimuth β and solar altitude α values are compared with the user's vision angle in the horizontal and vertical plane, respectively. In this way, it is verified whether or not

sunlight is shining into the road user vision field. Therefore, those hours in which sunlight does not shine into the user's vision are discarded.

In order to analyze whether the solar azimuth angle β coincides with the user's vision field in the horizontal plane, it is necessary to know the user's bearing with respect to the cardinal points, since the solar azimuth β is measured with respect to the south. The user's bearing is obtained from the vehicle trajectory; it is assumed as the MLS trajectory. The vehicle trajectory $T(T_x, T_y, T_z)$ is segmented with a spatial frequency σ, in order to obtain the driver's bearing vectors. Furthermore, locations $L(L_x, L_y, L_z)$ are obtained from the vehicle trajectory, over which the visibility of the driver is studied. Sun rays are thus defined as $θ(L_x, L_y, L_z, α, β)$. Specifically, $L_z$ is assumed to be the altitude of the road plus 1.1 m of the driver's height [17].

To examine whether the solar altitude angle α coincides with the user's vision field in the vertical plane, the slope of the road is also considered. Similar to the user's bearing, the slope of the road is obtained from the trajectory. If there is solar incidence in the vertical and horizontal plane (altitude and azimuth), the Sun can influence the user's vision field in cases where there are no intersections with obstacles. Figure 4 shows the sun incidence over the vehicle trajectory in both planes.

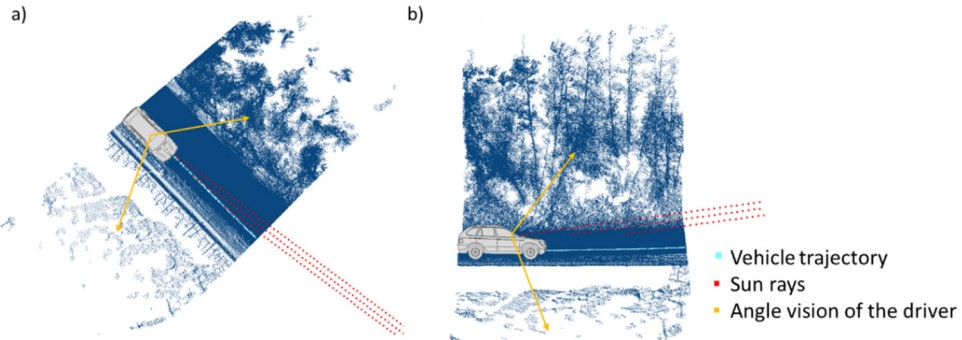

**Figure 4.** Solar incidence: (**a**) horizontal plane and (**b**) vertical plane.

### 3.3. Intersections with Near Road Obstacles

In the subsequent stage, we studied whether there are any near obstacles (vegetation, buildings, road elements, embankments) during the hours when sunlight intersects with the user's vision that could occlude the sunlight. The workflow related to this part of the methodology is shown in Figure 5.

MLS point cloud $M(M_x, M_y, M_z)$ is used as the input data. Before searching the intersections, the point cloud is segmented and triangulated. The trajectory of the road $T(T_x, T_y, T_z)$ is segmented in regular cross sections following the method proposed in [28]. Thus, director vectors of the trajectory are obtained for each segment. The point cloud $P(P_x, P_y, P_z)$ is segmented transversely based on the trajectory segments. Due to the laser range, the MLS point cloud contains outliers, distant points with noise not useful in the proposed method. Those points are eliminated with a statistical outlier removal (SOR) filter [29]. Another characteristic of MLS point clouds is the high point density near the road. This high point density is redundant for the proposed method, as well as increasing the computational time of future operations, including generation of an excessive number of triangles. Therefore, a point density reduction is applied based on a voxel structuring [30] with a voxel size $V_M$.

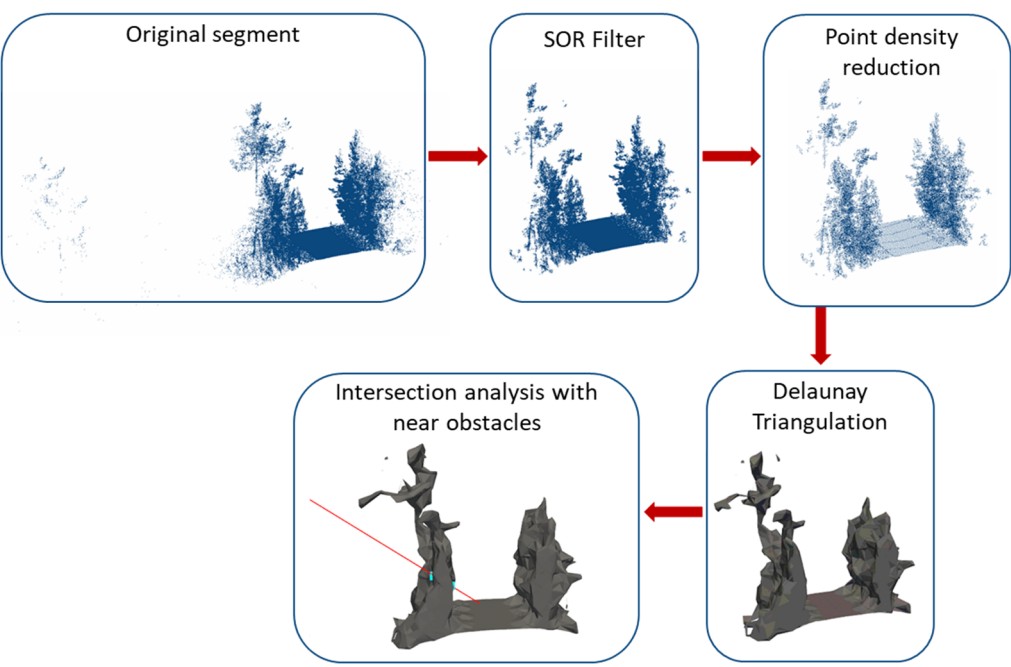

**Figure 5.** Workflow of intersections with near road obstacles.

After the MLS point cloud segmenting, cleaning and density reduction, a Delaunay triangulation is performed in order to compute the intersections of sun rays with near obstacles. A triangle mesh is generated to avoid the loss of intersections when sun rays pass through a gap in the point cloud. A Delaunay triangulation is a triangle mesh in which every triangle satisfies the Delaunay condition. This condition states that the circumcircle of a triangle includes only the vertex of the triangle, with the vertices being the points in the point cloud. To perform the Delaunay triangulation algorithm, the radius ρ of the circumference is chosen based on the distances between neighboring points. The smaller the radius ρ is, the less size of the triangles forming the mesh. The Delaunay triangulation connects the nearest neighbours; hence it can be used to model the intersection detection problem [31]. Then, intersections of each sun ray $\theta(L_x, L_y, L_z, \alpha, \beta)$ with the triangulated point cloud are sought.

### 3.4. Intersections with Distant Road Obstacles

Similar to the MLS point cloud processing, in the subsequent stage, ALS point cloud is analyzed whether or not sun rays intersect with any distant obstacles and the driver's vision. The maximum distance to be considered with respect to the study road is examined. An approach is proposed where the distance of the furthest obstacle that the user can see is chosen. The workflow related to this processing is shown in Figure 6.

First, ALS $A(A_x, A_y, A_z)$ data can contain scanned outlier points because of dust and false reflections, which are not integrated in the terrain. An SOR filter is applied in order to eliminate those points. Then, as points belonging to the road environment are analyzed in more detail with MLS data, the corresponding points are eliminated from the ALS data. For this purpose, a convex hull [32] is performed on the road segments previously obtained. ALS points within each convex hull polygon are removed. The point density in ALS is lower than in MLS but it is more constant; there are no variations with the distance to the road. However, for study visibility in very far elements to the road, such a level of detail is not necessary. A point density reduction decreases the computing time of the following processes. The voxel structure with size $V_A$ is employed to reduce the point density of the ALS data considering the distance of the road. Then, the processed ALS point cloud is converted into a triangle mesh utilizing the Delaunay triangulation. The radius ρ of the

triangulation varies depending on point density and, therefore, the distance to the road. Finally, intersections of sun rays $\theta(L_x, L_y, L_z, \alpha, \beta)$ with the mesh are estimated.

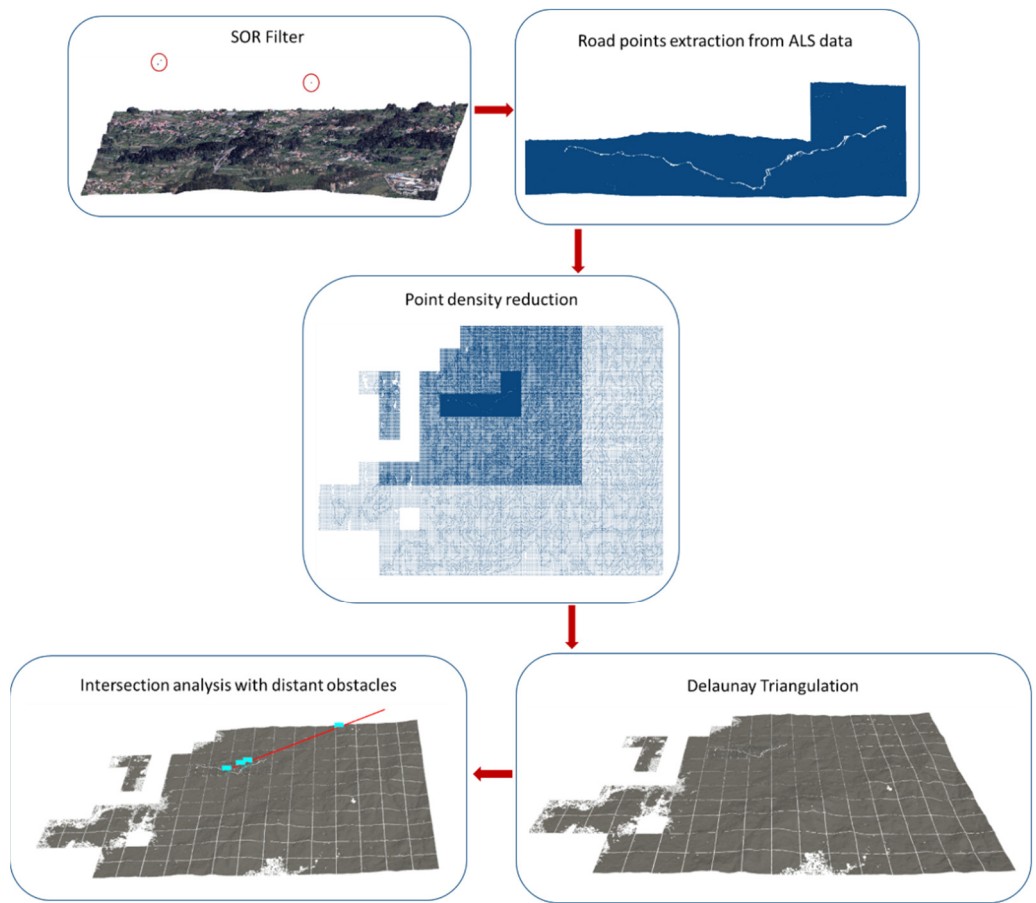

**Figure 6.** Workflow of intersections with distant road obstacles.

## 4. Experiments

### 4.1. Dataset

The proposed methodology was evaluated in three case studies (EP9701, EP9703, and EP2005) sited in Galicia (Spain), with the three corresponding MLS point clouds, acquired by LYNX Mobile Mapper of Optech [33]. ALS data was obtained from the Spanish National Geographic Information Centre (CNIG), considering 14 km as maximum distance to each road. The Spanish ALS point cloud database is organized in grids of 2 × 2 km. The number of grids analyzed in the north direction is reduced to those at a distance of 6 km. When sun rays are the most northerly, it is not required to consider grids in the north direction at a greater distance.

### 4.2. Parameters

Regarding point clouds data processing, the first step was to perform an SOR filter. Parameters used in MLS and ALS data were obtained empirically (Table 1).

**Table 1.** SOR filter parameters.

| Point Cloud | No. Neighbors | Standard Deviation |
| --- | --- | --- |
| MLS | 50 | 1 m |
| ALS | 20 | 5 m |

The parameters of point cloud density reduction are shown in Table 2. In MLS data, the voxel size $V_M$ was constant for all the roads, reducing the number of points by approximately 12%. In ALS data, the voxel size $V_A$ depends on the distance of the grid from the road. The closer to the road the smaller the voxel size value will be. Taking advantage of the grid distribution of the CNIG, the voxel size $V_A$ was selected according to the grid distance to the road. Related to the point density, the Delaunay triangulation depends on the radius parameter ρ. With smaller ρ, the triangulation is more detailed. However, it should be noted that in vegetation areas, a very small ρ radius can lead to excessive gaps in the triangulation mesh, since the vegetation points are more dispersed. In addition, the interest lies in obtaining the terrain profile; thus, a high accuracy is not required. Therefore, ρ was experimentally evaluated to obtain accurate results without generating numerous gaps in the mesh.

**Table 2.** Voxel size $V_M$, $V_A$, and ρ radius parameters.

| Point Cloud | Voxel Size | ρ |
|---|---|---|
| MLS | 0.2 m | 1 m |
| ALS → grids which contain the road | 20 m | 15 m |
| ALS → 2–6 km distance from the grid to the road-containing grid | 40 m | 40 m |
| ALS → 8–14 km distance from the grid to the road-containing grid | 80 m | 80 m |

*4.3. Results and Analysis*

4.3.1. Sun Trajectory Analysis for One Location

The sunrise and sunset time in the study area during the seasons were calculated (Table 3) in order to analyze the daylight hours, although they can be assessed at any day of the year. Road longitude is negligible compared with the Earth reference. Consequently, the difference in latitude and longitude along the road is also negligible. The difference between sunrise and sunset times at two different locations along the road are seconds. Therefore, the midpoint of the road was selected as the reference for calculating the sunrise and sunset times. Sun rays were obtained for the seasons regarding the period of daytime (Figure 7).

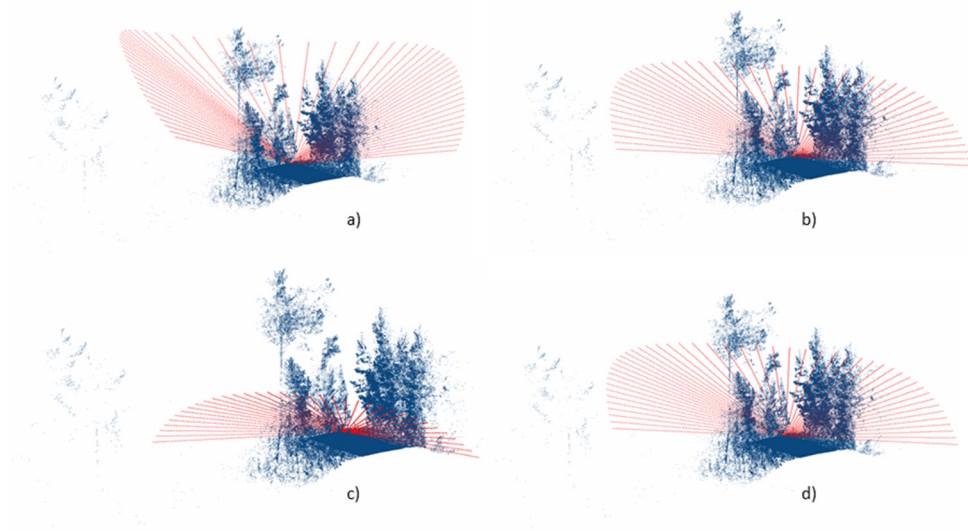

**Figure 7.** Sun trajectories during the daytime: (**a**) summer solstice, (**b**) autumn equinox, (**c**) winter solstice, and (**d**) spring equinox.

**Table 3.** Sunrise and sunset times during the seasons.

| Season | Sunrise Time | | Sunset Time | |
|---|---|---|---|---|
| | **Clock Time** | **Solar Hour** | **Clock Time** | **Solar Hour** |
| Summer solstice | 7:02 | 4:26 | 22:09 | 19:33 |
| Autumn equinox | 8:28 | 6:00 | 20:27 | 18:00 |
| Winter solstice | 9:06 | 7:33 | 17:59 | 16:26 |
| Spring equinox | 7:42 | 6:00 | 19:42 | 18:00 |

4.3.2. Analysis of Sun Glare Detections

The first case of study evaluated was the EP9701 road (Figure 8) located in the southwest of Galicia, with latitude 42.55° N and longitude 8.76° W. Specifically, the proposed method was examined in a 100 m section of this road which contained 2,111,823 points. It was surrounded principally by vegetation and had a descendant slope regarding south direction. The direction facing southeast was studied, corresponding to the incidence orientation of sun rays. Intersection results obtained for equinoxes and solstices are shown in Figure 9. The spatial frequency σ was 10 m (also for the other roads). The cells represent the time and segment (location) where sun rays had no incidence in the user's vision field; where sunlight was occluded by near obstacles, distant obstacles or both; and where sunlight had incidence in the road user's vision.

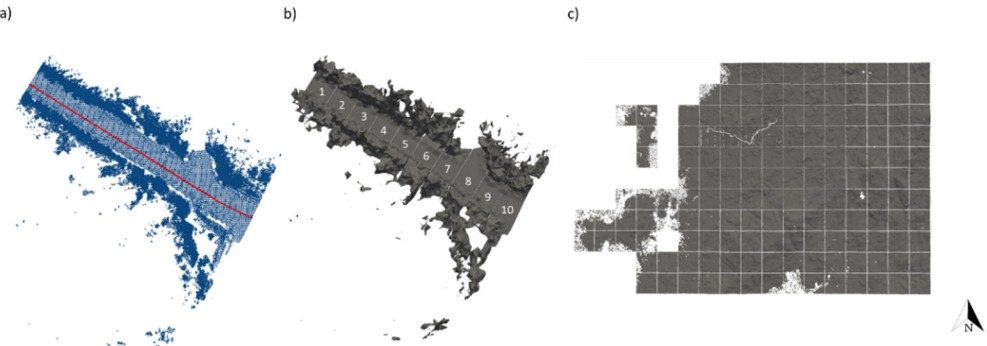

**Figure 8.** EP9701 road: (**a**) point cloud with vehicle trajectory, (**b**) MLS triangle mesh segments, and (**c**) ALS triangle mesh.

Regarding the road user's vision intersection with sun rays, the range that had solar incidence with the road user's vision did not go beyond the 12:00 solar hour because of the road bearing. The direct horizontal incidence over the user's vision was incremented towards the summer solstice (top view in Figure 10). However, considering the vertical vision plane, in summer the direct incidence was reduced by higher solar altitude α.

Intersections with near obstacles were mainly caused by trees. Figure 11 shows intersections for the location 65 m between sunrise and sunset. During winter solstice, intersections only occurred with the vegetation sited in the west due to the vehicle bearing and the period of the year. During equinoxes and summer solstice, intersections occurred in both sides of the road. Due to the ascendant slope of the vehicle trajectory, intersections in the late morning hours occurred with the road during the summer solstice.

Analyzing sun rays along the 100 m segment for the same hour, orientation and altitude of sun rays were different in each season, therefore, intersections did not occur in the same locations (Figure 12). During the winter solstice, there were intersections on the west side of the road, during the equinoxes there were no intersections, and during summer solstice, the intersections occurred on the east side of the road.

Intersections with distant obstacles were also analyzed for the different days. In this case study, most of the intersections occurred in the grids closer to the road. Figure 13 shows an example of intersections in the location 65 m during the summer solstice. Many distant

intersections coincided with the intersections caused by near obstacles, independently of the date.

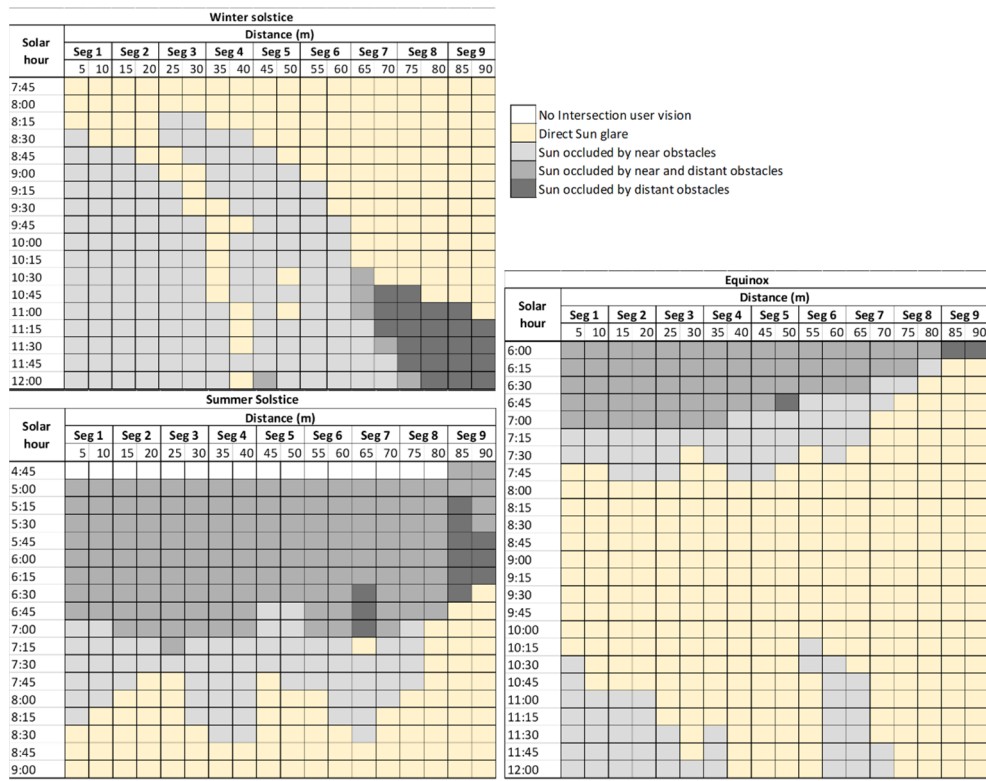

**Figure 9.** Intersections in EP9701 road during winter solstice, equinoxes, and summer solstice.

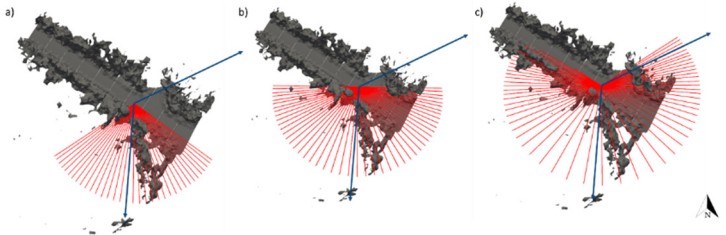

**Figure 10.** Sun rays in horizontal vision plane in EP9701 road: (**a**) winter solstice, (**b**) equinoxes, and (**c**) summer solstice.

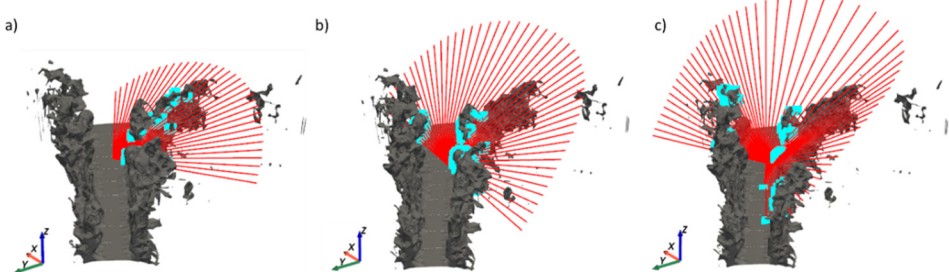

**Figure 11.** Intersections (light blue points) with near obstacles at 65 m: (**a**) winter solstice, (**b**) equinoxes, (**c**) summer solstice.

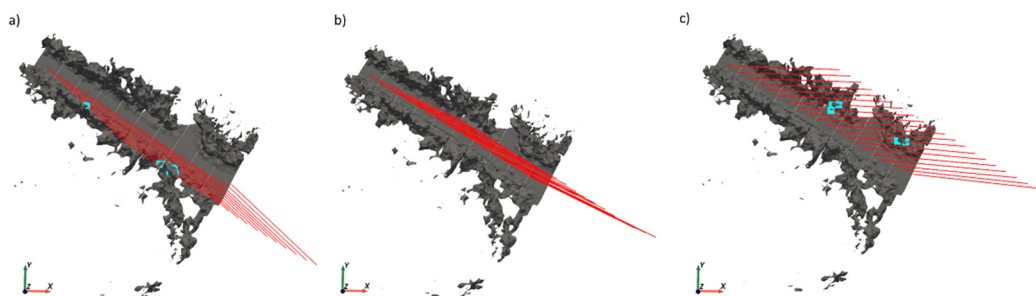

**Figure 12.** Intersections (light blue points) with near obstacles at 8:30 solar hour: (**a**) winter solstice, (**b**) equinoxes, and (**c**) summer solstice.

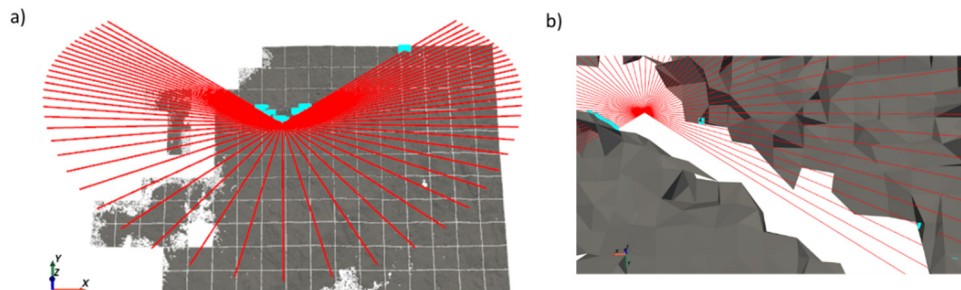

**Figure 13.** Intersections (light blue points) with distant obstacles at location 65 m in summer solstice: (**a**) complete ALS point cloud and (**b**) zoom near the analyzed location.

The second case study was performed in the EP9703 road (Figure 14) located in the southwest of Galicia, with latitude 42.56° N and longitude 8.80° W. The method was evaluated in a 100 m section of this road which contains 1,555,457 points. It was surrounded mainly by houses and low vegetation and had a descendant slope in a southerly direction. The direction facing the south was chosen to evaluate the method due to the Sun rising in the east and setting in the west. Figure 15 shows the range of time where sun rays had incidence in the user's vision as well as intersections with near and distant obstacles. During winter, the further east the vehicle trajectory was oriented, the earlier sunlight ceased to influence the user's vision. During the equinoxes, when the vehicle trajectory was oriented towards the west, sunlight did not influence the user's vision in the first daylight hours, however, when the segments were oriented more towards the east, sunlight did not influence the user's vision in the late daylight hours. The incidence of sunlight on the user's vision during the summer solstice occurred in the last four segments, which were oriented to the east. In the first segments, the vehicle trajectory was oriented to the south; therefore, sun rays had incidence in the middle hours referring to the horizontal plane. At these hours, the solar altitude $\alpha$ was too high and it did not intersect with the road user's vision.

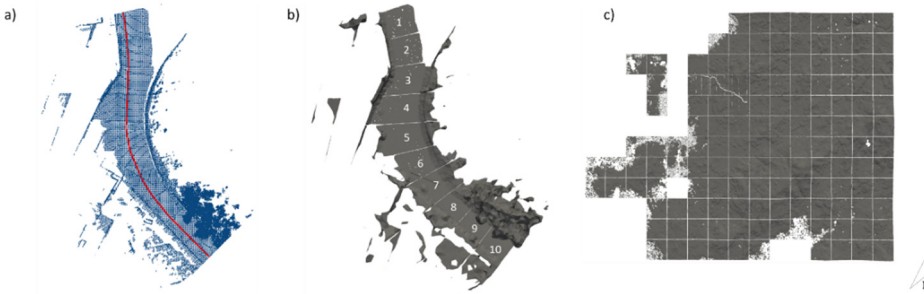

**Figure 14.** EP9703 road: (**a**) point cloud with vehicle trajectory, (**b**) MLS triangle mesh segments, and (**c**) ALS triangle mesh.

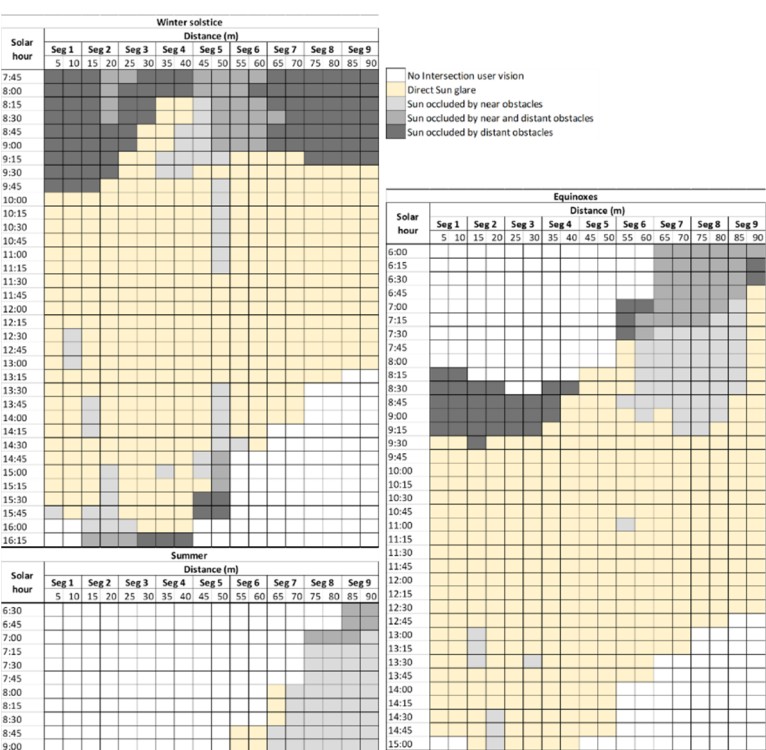

**Figure 15.** Intersections in EP9703 road during winter solstice, equinoxes, and summer solstice.

Intersections with near obstacles were analyzed regarding different dates. Figure 16a,b shows the influence of the solar altitude α in intersections between seasons. Intersections with distant obstacles were also studied. An example of intersections in the location 50 m during winter solstice can be seen in Figure 16c. During the summer and equinoxes, although there were intersections with distant obstacles in the early hours of the day, they were not considered since sunlight did not affect the user's vision.

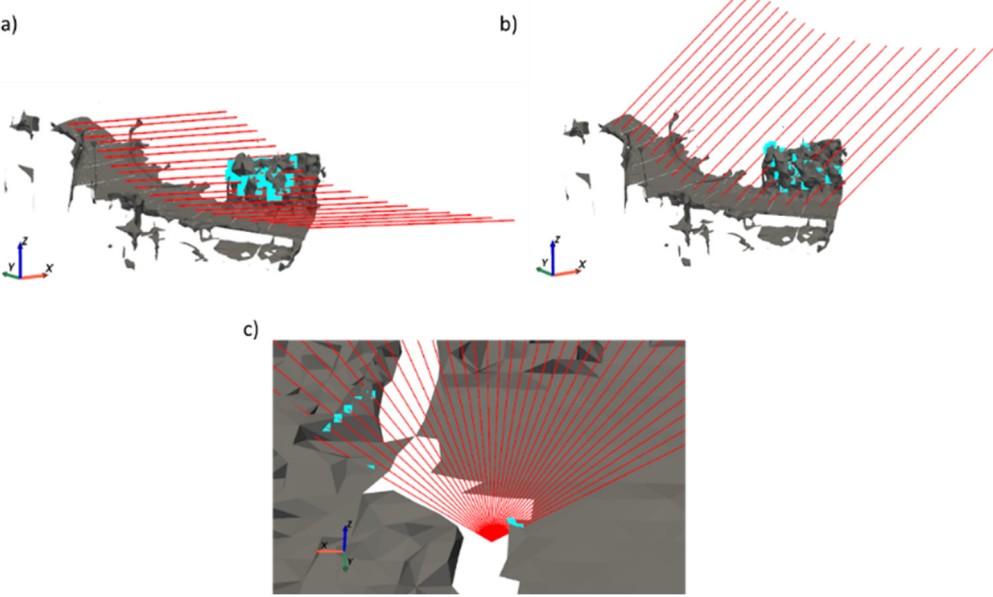

**Figure 16.** Intersections (blue points) with obstacles: (**a**) intersections with near obstacles at 9:00 solar hour in winter solstice, (**b**) intersections with near obstacles at 9:00 solar hour in summer solstice, and (**c**) zoomed view of intersections with distant obstacles at 50 m.

### 4.3.3. Analysis of Sun Ray Intersections with the Mesh

The quality of sun ray intersections depended on the Delaunay triangulation. Most of the triangles were generated correctly, however, in the ALS data, some empty spaces were generated. Intersections of sun rays $\theta(L_x, L_y, L_z, \alpha, \beta)$ with near and distant obstacles were also tested, verifying that no intersections were lost. In the MLS data, linear elements (wires) were also polygonized; therefore, intersections with sun rays were generated. These intersections, although they were well estimated, were too small and the sunlight was not occluded. In elements with a point distribution adjusted to a surface, as in vegetation, many triangles overlapped and were generated inside the vegetation surface. This was not an optimal solution, however, it did not imply a loss of intersections.

## 5. Discussion

The contribution of the proposed method consists of the detection of sunlight intersections caused by both near and distant obstacles present in the environment and calculated from MLS and ALS point clouds, based on the trajectory data given the relevance of the driver's bearing and slope on the solar incidence. The method can be carried out for a given location for any day of the year, even though only the direct sun glare has been taken into consideration. Calculations are made for sunny days. The existence of clouds or other phenomena minimize the existence of glare; however, meteorology is outside the scope of this work. During adverse weather conditions, such as fog, rain, or snow, the solar incidence is different due to reflections of sunlight. Therefore, in these cases, the solar incidence should be recalculated.

In order to analyze the intersections of sun rays $\theta (L_x, L_y, L_z, \alpha, \beta)$ with near obstacles, several tests were performed varying the spatial frequency $\sigma$ in 1 m, 5 m, and 10 m; therefore, the distance between sun rays also varied (Figure 17). Intersections in winter solstice considering the vision angle of the road user and near and distant obstacles are shown in Figure 18.

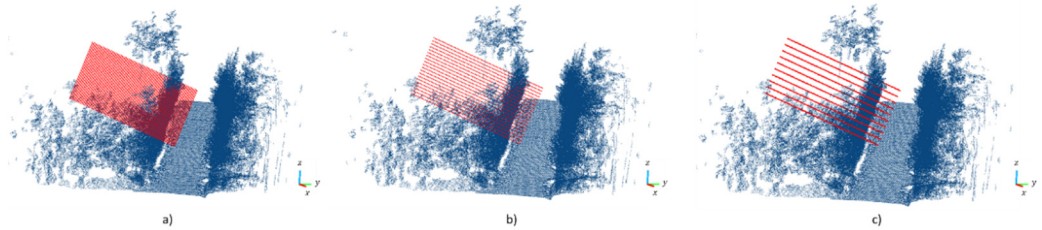

**Figure 17.** Spatial frequency $\sigma$ of sun rays: (**a**) 1 m, (**b**) 5 m, and (**c**) 10 m.

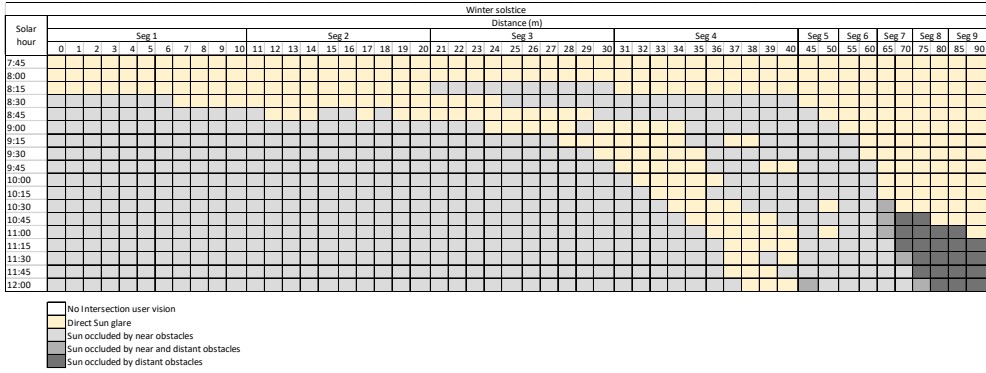

**Figure 18.** Intersections in road EP9701 during winter solstice.

The results demonstrate that it is not necessary to select 1 m as the sun ray displacement distance along the segment, as in general there were no intersection differences with this displacement distance. Results choosing 10 m as the sun ray displacement distance reveal

that there was information that can be lost, as in the case of segment 4, between 30 m and 40 m within 10:00 and 10:45 solar hours. If this distance was chosen, some direct sun glares or shadows were lost. Another case occurred during segments 2–4 at 8:15 solar hour. If 10 m displacement was selected, it was assumed that from 20 m to 30 m there was sun glare and between 30 m and 40 m there were obstacles that occluded the sunlight. Consequently, information was lost during a large range. This information was preserved by choosing 5 m as the sun ray displacement. With the spatial frequency σ of 5 m, occasional reflections were lost, although they did not represent a constant glare. Therefore, 5 m was selected as the distance for the study of the intersections along the road.

Combining MLS point clouds and ALS point clouds provides more level of detail about the elements around the road. There were areas where the sunlight was apparently incident because it did not intersect with any near obstacle. However, ALS point clouds provided additional information, proving that the sunlight did not shine directly due to the terrain's altitude. Obstacles in the environment were represented by surfaces according to the Delaunay triangulation, which led to a higher probability to find intersections. With the MLS triangulation, vegetation, houses, and other near obstacles were clearly represented (Figure 19a). Figure 19b,c, show the ALS 100 m segment triangulation with ρ = 4 and ρ = 8, respectively. In both, it can be seen that the road and surroundings were not clearly defined, due to the point density being lower than in the MLS data. Consequently, MLS data are more appropriate for calculating intersections with near obstacles. MLS input data may have some limitations due to data collection. In some cases, dynamic objects such as cars or trucks could be acquired and calculated as influencing intersections. In the case of ALS, altitude changes in the terrain were well defined, with more detail in the grids containing the road and less detail as the grids become more distant from the road (Figure 19d).

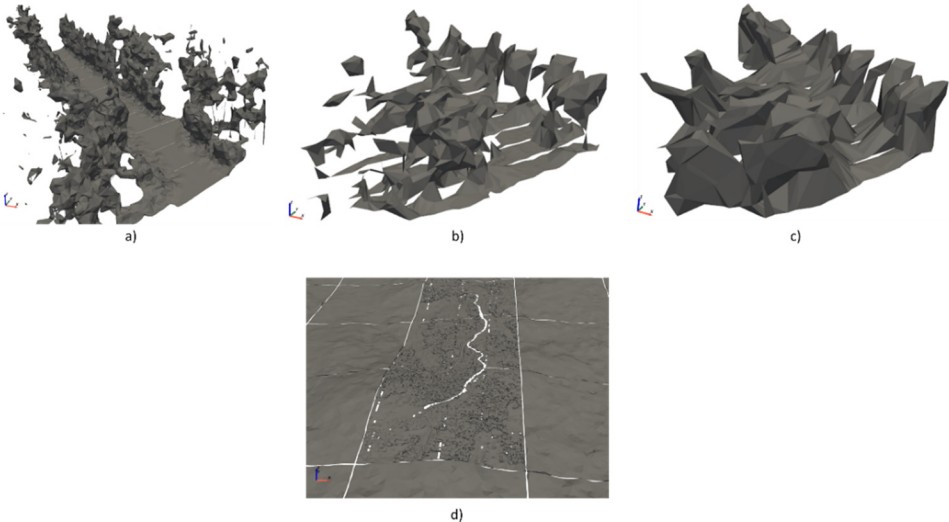

**Figure 19.** Delaunay triangulation: (**a**) MLS data, (**b**) ALS 100 m segment ρ = 4, (**c**) ALS 100 m segment ρ = 8, and (**d**) ALS data.

*Processing Time*

The algorithm was implemented in Python and processed on an Intel Core i7 CPU 3.40 GHz with 16GB RAM. The processing time was 2 min 48 s to perform the road trajectory and MLS point cloud segmentation in 100 m segments, considering 1 km of the road. The mean processing time to calculate sun rays and obtain its incidence on the user's vision in one location was 9 s.

The processing time is detailed in Table 4 considering that the calculations were performed during the summer solstice. A sun ray was calculated every 15 min. In this season, more sun rays were generated (61) due to the longer daylight hours. It should be noted that convex hull was done in each segment of 10 m along the case studies' roads, in which the total length was about 10 km and 150,500,000 points. ALS data processing required higher time

consumption due to the number of points. However, the Delaunay triangulation was more time consuming with MLS data because the ρ radius chosen was smaller and the proportion of point density related with the size of the scanned area was approximately 60 times higher than in ALS data, after applying the SOR filter and point density reduction. This process could be replaced in future work with a surfel-based method in order to evaluate the processing time. Point cloud segmenting, cleaning, density reduction, and Delaunay triangulation code were executed once, due to the data were stored. Intersections search code was executed several times in order to perform different tests.

**Table 4.** Processing Time Detail.

| Operation | No. | Processing Time |
|---|---|---|
| Trajectory and MLS segmentation (1 km → **100 m segments)** | **16,500,000 pts** | **2 min 48 s** |
| **Sun ray calculation and its incidence in user's vision (one location)** | **61 rays** | **9 s** |
| **MLS data processing (100 m)** | | **37 s** |
| SOR and point density reduction | 230,000 pts | 8 s |
| Delaunay triangulation | 150,000 triangles | 20 s |
| Intersections with near obstacles (one location) | 61 rays | 9 s |
| **ALS data processing (1 grid with road)** | | **8 min 53 s** |
| SOR, convex hull and point density reduction | 3,500,000 pts | 7 min 29 s |
| Delaunay triangulation | 250,000 triangles | 1 s |
| Intersections with distant obstacles (one location) | 61 rays | 1 min 23 s |
| **ALS data processing (1 grid with surroundings)** | | **1 min 36.2 s** |
| SOR and point density reduction | 3,500,000 pts | 13 s |
| Delaunay triangulation | 5,000–90,000 triangles | 0.2 s |
| Intersections with distant obstacles (one location) | 61 rays | 1 min 23 s |

## 6. Conclusions

Mapping the areas of the road where sun glare occur is key to analyzing accidents and proposing measures to reduce them. In this paper, a new method for detecting sun glare in roads is presented. The method calculates the direct incidence of sun rays, considering the Sun's position, driver position, driver bearing, road slope, and searching for intersections with near and distant obstacles in the road environment from MLS and ALS point clouds. The proposed method is configurable for use at different times and days of the year, as well as for different intervals along a road in any location.

The experimental results, on two different roads located in Galicia (Spain), show a correct detection of sun glares and shadows caused by the intersection of sun rays with the triangulation from the point clouds. The spatial frequency was a very important parameter between precision of sunny/shadow areas and computational time. The selected displacement of the trajectory provides accurate intersections with near and distant obstacles, with almost no loss of information. It was considered that short exposure sun glare, although annoying, is not as dangerous as a prolonged exposure over many meters. In the case studies, it was observed that, despite being roads with a high number of surrounding objects (houses and vegetation), there was a large exposition of the driver to direct sun glare. The results clearly indicate the time range in which sunlight is incident on the user's vision during different dates, which depends on the bearing of the road. The further east the road faces, the more solar incidence will affect drivers in the early hours. The further west the road is, the more solar incidence will affect drivers in the late daylight hours. Solar altitude $\alpha$ is also a significant feature. During the central hours of the summer solstice, sun rays are not incident on users' vision because the solar altitude $\alpha$ is too high. By contrast, during the winter solstice, the Sun is at its lowest position and in general, this is when sunlight has the greatest influence on users' vision. The combination of MLS and ALS data proved to be successful. The MLS data were acquired with a linear perspective very

similar to the driver's vision and provided a very high level of detail even for identifying intersections of sun rays with wires. The ALS data had insufficient point density for a correct visibility analysis of near obstacles; however, it had sufficient density to analyze large distant terrain obstacles not acquired with MLS. The results obtained can be used for the implementation of road signs or for warnings in GPS navigation systems. They can also be used to study the placement of elements around the road that occlude sunlight, for example, by adding vegetation.

In future work, the incidence of the indirect sun glare will be studied, considering days with adverse weather conditions (rain, snow, fog) and analyzing sunlight reflections. For this purpose, a photon tracing and mapping algorithm can be use in order to simulate the reflections of sunlight with the environment. In addition, the identification of the objects that generate each shadow will be studied, as well as the integration of natural or artificial elements that minimize drivers' exposure to sunlight will be proposed. The visual space is represented in this method by a horizontal and a vertical plane. A cone will be modeled to simulate the visual space of a human or of a camera to detect sun glare.

**Author Contributions:** Conceptualization, E.G., J.B. and S.M.G.-C.; methodology and software, S.M.G.-C.; validation, J.B., S.M.G.-C. and P.d.R.-B.; investigation, E.G.; resources, E.G.; writing, S.M.G.-C.; writing—review and editing, E.G. and J.B.; visualization, J.B., S.M.G.-C. and P.d.R.-B.; supervision, E.G. and J.B. All authors have read and agreed to the published version of the manuscript.

**Funding:** This research was funded by the Xunta de Galicia, grant numbers ED481B-2019-061 and ED431C 2020/01, and by the Ministerio de Ciencia, Innovación y Universidades -Gobierno de España-, grant numbers PID2019-105221RB-C43/AEI/10.13039/501100011033 and TIN2016-77158-C4-2-R. This paper was carried out in the framework of the InfraROB project (Maintaining integrity, performance and safety of the road infrastructure through autonomous robotized solutions and modularization), which has received funding from the European Union's Horizon 2020 research and innovation programme under grant agreement no. 955337. It reflects only the authors' views. Neither the European Climate, Infrastructure, and Environment Executive Agency (CINEA) nor the European Commission is in any way responsible for any use that may be made of the information it contains.

**Institutional Review Board Statement:** Not applicable.

**Informed Consent Statement:** Not applicable.

**Data Availability Statement:** Not applicable.

**Acknowledgments:** The authors would like to thank those responsible for financing this research.

**Conflicts of Interest:** The authors declare no conflict of interest.

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
