# Peer review of "Detection of Direct Sun Glare on Drivers from Point Clouds"

_remotesensing, doi:10.3390/rs14061456_

Round 1

Reviewer 1 Report

Dear authors thank you for the interesting article. The text is interesting and well described, with a clear explanation and narration easy to understand for all readers. The analyzed topic is important and almost all car drivers experience blinding sun glares on sunny days. 

Dear authors, please clarify the following issues:
 - in my opinion, the number of pages in this article is too high. The ordinary reader will lose their focus somewhere in the middle of the text. This is a perfect chapter for a monography, rather than an article - the number of pages should be reduced. Please reduce the number of case studies - all are the same and there are no significant differences between them. 
- as was suggested in the text this approach may be applied for autonomous cars and in a similar approach where the camera may be exposed to direct sun glare. But to implement this approach all roads should be 3D scanned and analyzed (even if this approach will base on the available data the topic is challenging). If we look in the welding industry similar problem was solved by applying the auto-darkening welding helmets. Please comment on the practical implementation of your method - is it possible? In my opinion, a few sentences should be included in the text to explain how you plan to use the results of your work. 
- has the proposed method been verified? The key aspect is the accuracy of the mapping of the real environment, on the other hand, too high accuracy makes it impossible to quickly interpret the results - so a compromise is needed. 
- please correct the editing errors in th e following lines: 74, 106, 176,197, 220, 228, 231, 289, 293, 309, 315, 323, 328, 341, 348, 366, 372, 377, 397, 408, 424, 456, 458, 482, 490.
- section "3.1 Sun position calculation" is unnecessary detailed,
- the size of the symbols in figures 2, 3 should be increased to improve readability.

Author Response

The responses to the reviewer's comments are detail in the word file "Response to Reviewer 1"

Reviewer 2 Report

Detection of direct sun glares on drivers from point clouds

The paper proposess sun glare detection based on point clouds from Aerial Laser  Scanning and Mobile Laser Scanning. The point clouds are filtered and Delaunay-triangulated. Sun rays are intersected with the triangulation and sun glares are  calculated for a driver's trajectory. Examples obtained from different data sets  and sun positions are presented.

The algorithmic design is rather straight-forward. There is neither an analytic justification, nor a comparison with respect to competing methods. Also, important  details, like the number of sun rays used for intersection and computation times are missing.

When considering the application, I would assume that photon tracing and mapping would be the method of choice, as it calculates a point-based representation of  reflected sun light that can be collected efficiently at varying driver's locations. Also, multiple reflections are simple to model. Unfortunately, this 
approach is not mentioned, at all.

Another point is the use of Delaunay triangulations. There is no justification given, why a triangulated representation would perform better than, for example, a representation due to surface elements (surfels, i.e. points with normal vectors). Surfel-based representations are easily obtained from point clouds and are well suited to photon mapping.

Further, the algorithm may be compared to competing methods [1] that should also be included:

[1] Xiaojiang Li, Bill Yang Cai, Waishan Qiu, Jinhua Zhao, Carlo Ratti:
A novel method for predicting and mapping the presence of sun glare using Google 
Street View. Computer Vision and Pattern Recognition abs/1808.04436 (2018)

[2] Keisuke Yoneda, Naoki Ichihara, Hotsuyuki Kawanishi, Tadashi Okuno, Lu Cao, 
Naoki Suganuma: Sun-Glare region recognition using Visual explanations for Traffic 
light detection. IEEE Intelligent Vehicles Symposium (IV) 2021: 1464-1469

the draft contains erratic cross-references: "Error! Reference source not found"

Author Response

The responses to the reviewer's comments are detail in the word file "Response to Reviewer 2". Please see the attachement.

Round 2

Reviewer 1 Report

Dear authors, thank you for considering my suggestions and applying the suggested changes.

After a second reading of the text a few additional suggestions appears:
- visual layer,
-- figure 3 and 4 - please normalize text size in the description of the pictures according to description size in other pictures
-- figure 5 and 6 part of the blocks has cut border,
- description
-- in the text added new point 4.3.4, in fact, this is not "time processing" is rather "processing time" - please check if I good understand your intentions correctly (also in the text, name, and in the description of table 4).
-- added new table 4 brings interesting information - please consider adding even one sentence in the discussion part, which discusses the data presented in the table. 

Reviewer 2 Report

The paper has improved in this revision by adding explanation and table 4 with computation times. Unfortunately, the table does not contain the number of primitives generated (points, triangles, rays, etc.). This can differ significantly for multiple data sets. Please, provide all numbers for the examples shown and discuss how the computation times behave asymptotically with growing data complexity.
